# Analysis and Simulation of Glioblastoma Cell Lines-Derived Extracellular Vesicles Metabolome

**DOI:** 10.3390/metabo10030088

**Published:** 2020-03-02

**Authors:** Miroslava Čuperlović-Culf, Nam H. Khieu, Anuradha Surendra, Melissa Hewitt, Claudie Charlebois, Jagdeep K. Sandhu

**Affiliations:** 1Digital Technologies Research Centre, Bldg-M50, 1200 Montreal Road, National Research Council Canada, Ottawa, ON K1A0R6, Canada; Anuradha.Surendra@nrc-cnrc.gc.ca; 2Human Health Therapeutics Research Centre, Bldg-M54, 1200 Montreal Road, National Research Council Canada, Ottawa, ON K1A0R6, Canada; NamHuan.Khieu@nrc-cnrc.gc.ca (N.H.K.); Melissa.Hewitt@nrc-cnrc.gc.ca (M.H.); Claudie.Charlebois@nrc-cnrc.gc.ca (C.C.)

**Keywords:** extracellular vesicles, metabolomics, metabolism modeling, machine learning, glioblastoma

## Abstract

Glioblastoma (GBM) is one of the most aggressive cancers of the central nervous system. Despite current advances in non-invasive imaging and the advent of novel therapeutic modalities, patient survival remains very low. There is a critical need for the development of effective biomarkers for GBM diagnosis and therapeutic monitoring. Extracellular vesicles (EVs) produced by GBM tumors have been shown to play an important role in cellular communication and modulation of the tumor microenvironment. As GBM-derived EVs contain specific “molecular signatures” of their parental cells and are able to transmigrate across the blood–brain barrier into biofluids such as the blood and cerebrospinal fluid (CSF), they are considered as a valuable source of potential diagnostic biomarkers. Given the relatively harsh extracellular environment of blood and CSF, EVs have to endure and adapt to different conditions. The ability of EVs to adjust and function depends on their lipid bilayer, metabolic content and enzymes and transport proteins. The knowledge of EVs metabolic characteristics and adaptability is essential for their utilization as diagnostic and therapeutic tools. The main aim of this study was to determine the metabolome of small EVs or exosomes derived from different GBM cells and compare to the metabolic profile of their parental cells using NMR spectroscopy. In addition, a possible flux of metabolic processes in GBM-derived EVs was simulated using constraint-based modeling from published proteomics information. Our results showed a clear difference between the metabolic profiles of GBM cells, EVs and media. Machine learning analysis of EV metabolomics, as well as flux simulation, supports the notion of active metabolism within EVs, including enzymatic reactions and the transfer of metabolites through the EV membrane. These results are discussed in the context of novel GBM diagnostics and therapeutic monitoring.

## 1. Introduction

Glioblastoma (GBM) is the most aggressive and malignant grade IV astrocytoma of the central nervous system. Currently, GBM diagnosis generally translates into a very poor prognosis with limited treatment options and an average patient survival of only 14–16 months [1]. Diagnosis of GBM requires magnetic resonance imaging (MRI) and validation by an invasive intracranial biopsy with treatment modalities including surgical resection, radiation and chemotherapy, followed by serial MRI scans of the brain in order to detect tumour cell infiltration into normal tissue. Apart from logistics and cost issues with serial MRI analysis, it is also very difficult and challenging to distinguish between tumour- and treatment-related effects. Image analysis is becoming increasingly more automated and accurate with the introduction of machine learning and particularly deep learning, reviewed recently by Zaharchuk et al. [2]. However, even if these methodologies are adopted for image analysis, there remain issues with MRI accessibility, cost and logistics. In spite of great advances in image analysis of GBMs, there is no FDA-approved biomarker for GBM diagnosis. Therefore, there remains an urgent need to develop non-invasive, reliable and easily measurable biomarkers for GBM diagnostics and theranostics.

The use of extracellular vesicles (EVs) is gaining momentum and producing promising results in the field of GBM diagnosis [3,4]. EVs are micro-sized (microvesicles, 100–1000 nm) and nano-sized (exosomes, 30–150 nm) vesicles that are actively released by almost all cell types and comprise a highly sophisticated intercellular communication system for exchanging biological information with adjacent or distal cells [5,6]. While microvesicles are released from direct budding of the plasma membrane, exosomes are produced via the multivesicular endosomal pathway and secreted into the extracellular mileu when they fuse with the plasma membrane [7,8]. As part of exosome biogenesis, distinct messages in the form of lipids, proteins, nucleic acids and metabolites are packaged into exosomes and delivered into recipient cells, leading to the modulation of a range of cellular function [5,6,7].

Accumulating evidence has shown that cancers, including GBMs, release large amounts of EVs, including exosomes, into the bloodstream, offering a new opportunity for non-invasive biomarker discovery. Circulating EVs from glioblastomas have been shown to contain microRNA [9,10] and DNA [11]. Several laboratories have published the proteomic profiles of GBM-derived EVs and identified specific proteins highly enriched in EVs [12,13,14,15]. Mallawaarachy et al., have studied six GBM cell lines and identified 844 proteins in EVs with 145 proteins common to all EVs [13]. Limited analyses of the lipid composition of EVs showed a very different content to parental cells with major enrichment in glycosphingolipids, sphingomyelin, cholesterol and phosphatidylserine [16]. To date, very little attention has been paid to the study of metabolites in EVs with only a limited number of published studies. Metabolites represent the intermediate or end products of cellular processes occurring downstream of genomic or proteomic regulation and can provide specific and sensitive information, making them a preferred diagnostic biomarker. For example, EVs derived from the urine of prostate cancer patients have been shown to be enriched with several cytosolic metabolites [17]. Amongst these enriched metabolites are the members of the nucleotide and spermidine pathways that can be linked to several enzymes or transporters within EVs [17]. The levels of glucoronate, D-ribose 5-phosphate and isobutyryl-L-carnitine were 2–26-fold lower in all pre-prostatectomy samples compared to the healthy control and post-prostatectomy samples. Although these analyses provide interesting diagnostic leads, the cohort sizes of these studies were very small, and therefore results need to be validated in a larger cohort in order to advance to future clinical applications.

Since EVs contain protein, nucleic acids and lipid cargoes that relate to the parental cells, they represent an extremely interesting diagnostic tool. Given the ability of EVs to transfer bioactive material to specific cells across biological barriers, they are also being exploited as potential drug delivery vessels. However, despite increasing efforts to determine the cargo and function of EVs, understanding of multiple roles played by EVs in tumour progression is still lacking. The possibility of using protein cargoes of EVs for diagnosing tumour subtypes, particularly for GBMs, has been suggested [15,18], while at the same time metabolomics have been used to determine GBM subtypes using metabolomics analysis of cells or media [19]. Metabolomics analysis of GBM-derived EVs can provide easily measurable non-invasive biomarkers for diagnosis as well as GBM subtype determination. At the same time, understanding the functional role of enzymes in EVs and the possibility of an active metabolism within EVs is essential for both the utilization of EV metabolome as a diagnostic tool and increasing understanding of the behavior and function of EVs in different environments.

Although there is a great deal of enthusiasm for the development of exosomes as reservoirs of biomarkers and novel drug delivery platforms, our understanding of the basic biology of exosomes remains very limited, constrained by the lack of appropriate experimental methods for the efficient separation and visualization of these nanoparticles. Computational simulations, combined with experimental work on exosomes, can provide a window into the behavior of these important nanovesicles in different conditions and the development of this novel diagnostic modality for diagnosing GBM, therapeutic monitoring, and assessing treatment resistance.

## 2. Results and Discussion

In this study, three types of typical malignant glioblastoma (GBM) cells, namely LN18, A172 and U118, were grown in complete medium (DMEM+10% FBS). Microscopic examination of these cultures revealed epithelial-like cells growing as monolayers with a doubling time of ~48 h. Cultures of normal human astrocytes (NHA) grew as a monolayer of large flat cells with a polygonal morphology with a doubling time of ~72 h. In an attempt to identify the metabolic cargo of GBM-EVs and NHA-EVs, GBM and NHA cells were grown for 72 h in complete medium-lacking serum. Both GBM and NHA cells retained the fibroblast-like morphology with no change in cell viability, assessed by trypan blue staining. The presence of CD9 is considered as a pan-exosomal marker and is shown to be expressed by small EVs. Accordingly, Western blot analysis of proteins from cell lysates and EVs probed with anti-CD9 antibody showed the presence of ~25 kDa band in cell lysates, with EVs showing a slightly lower band (Figure 1).

Metabolites were extracted from GBM cells (LN18, A172 and U118) and normal human astrocytes (NHA), GBM and NHA-derived EVs, and their respective media, as described in Materials and methods. NMR metabolomics analysis was performed for hydrophilic extracts (see Figure 10 for experimental workflow and sample size). Three GBM cell lines have been studied extensively in the past and are known to possess number of different characteristics including Phosphatase and tensin homolog (PTEN); EGFRvIII mutant; and CDKN2A, outlined in Table 1 (obtained from [20,21] and www.expasy.org).

One-dimensional (1D) 1D ^1^H Nuclear Overhauser Effect Spectroscopy (NOESY) NMR experiments were carried out on all extracts and complete spectra are available upon request. 1D proton spectra for all samples were processed and aligned using the Icoshift method [22], as described in Materials and Methods (spectra are shown in Appendix A). Principal component analysis (PCA) of complete 1D spectra illustrated variances in the metabolic profiles across all samples based on complete metabolic profiles in an untargeted, i.e., qualitative, sense (Figure 2A). In the PCA analysis of all samples, Principal component 1 (PC1) showed separation between cells, media and EV metabolic profiles, as well as metabolic profiles of GBM and NHA cells from all sample sources. Principal component 2 (PC2) revealed major differences between the metabolic profiles of cells and media on one side and EVs of GBM cell lines on the other side, with the additional separation of EVs obtained from U118 cells as one group and LN18 and A172 cells as another. Interestingly, PC2 of the cell and EV spectra shows separation between GBM subtypes with clear cell type separation provided by EV metabolic profiles. 

An alternative analysis approach on complete spectra—T-distributed stochastic neighbor-embedding (t-SNE) [23] (Figure 2B) showed similar sample separation. T-SNE has been shown to be particularly suitable for large datasets, where it retains the local structure of the data while also revealing some important global structures. 

T-SNE (Figure 2B) clearly shows separation by cell type and sample source with the separation of U118 EV spectra from LN18 and A172 EV results. Similarly to PCA, t-SNE based on EV metabolome shows the separation of U118 from LN18 and A172 cells more clearly then in either cell or media analysis. The difference between cell types based on profiles measured in cell, media and EV is presented by PCA in Appendix A. These results further present a major difference between metabolic profiles of media, cells and EVs in GBM lines and a smaller, relative separation between three sample groups in normal, NHA cell line. Metabolic profiles of EVs allow separation in PC1 of U118 from NHA and the other two GBM lines (A172 and LN18). The observed difference between GBM cell lines is in agreement with a previously published analysis of apoptosis in GBM cell lines clearly showing a difference between A172 and U118 cell lines (LN18 was not included in the previous study) [24]. 

In order to determine specific metabolic differences between cell line and sample type groups, we have performed assignment and quantification of relative metabolite concentrations using the methodology described previously [19,25] and briefly outlined in Materials and Methods. The resulting relative concentrations for the 50 identified metabolites following scaling to a mean of zero and standard deviation of 1 across metabolites and samples are graphically represented in a heat plot (Figure 3) following the hierarchical clustering of metabolites and samples for easier visualization.

Figure 3 illustrates the variability in metabolic profiles across cell lines and sample sources as well as biological replicates. Similarities in relative concentration change between cells, media and EVs in three GBM cell lines is apparent with the clustering of all GBM cells, media and EVs and a separate cluster of NHA samples, similar to the results of the PCA and t-SNE of spectra (Figure 2). For the majority of metabolites, there is a larger similarity between EV and media then EVs and cells. 

The determination of the metabolites with the most significant concentration difference between cells and EVs has been performed with ANOVA feature selection, provided in Orange [26]. ANOVA ranks features based on the difference between average values in different classes. The resulting metabolites with an ANOVA rank of over 5 are shown in Figure 4.

In all three GBM cell lines, the level of methionine is lower in EVs than in cells. With the known dependence of GBM cells on methionine [27], it can be expected that methionine will be retained in cells. In fact, in all cell lines explored in this work, methionine levels are significantly higher in cells than in EVs or media. As one of the essential amino acids, methionine has to be obtained from the media and has a major role as a precursor of other sulfur-containing amino acids (e.g., cysteine) and their derivatives (e.g., glutathione), which are all essential for cellular function. 

Unlike U118 cells, A172 and LN18 lines show a major concentration increase in glycerol, tryptophan, carnitine and oxidized glutathione (GSSG) in EVs relative to cells. Glycerol is a by-product and component of several metabolic pathways such as glycolysis, galactose metabolism, and a precursor for glycerolipid metabolism as well as a necessary metabolite for the formation of triglycerides and subsequent fatty acid biosynthesis, a process known to be highly activated, particularly in more aggressive GBMs [28]. Carnitine, once again, is over-concentrated in the EVs of LN18 and A172 and is also involved in fatty acid metabolism [29]. Furthermore, carnitine is a substrate for a major factor OCTN2 (SLC22A5) in energy metabolism, recently shown to be overexpressed in GBMs, particularly in more aggressive subtypes of GBM [30]. Tryptophan, together with arginine, prostaglandin and adenosine, has recently emerged as an immuno-metabolic node in some subtypes of GBM [31]. Therefore, the observed high concentration of tryptophan in EVs of A172 and LN18 cells could play a role in immune response. Finally, glutathione level in GBM cells has been strongly linked with drug resistance and its concentration is different between different GBM cell lines [32]. The LN18 line has been previously shown to over-express GSH-related metabolic enzymes [33] with a high antioxidant capacity. In our analysis, the level of reduced GSH is higher in A172 and LN18 cells than in corresponding EVs, while GSSH is over-concentrated in EVs, possibly as a way to remove the oxidized form of glutathione from the cells, thereby affecting the GSH/GSSH ratio. The reduction of glutathione is a major antioxidant system in cells, that has been shown to play an important role in tumor progression and treatment resistance [34]. The presence of gamma–glutamyltransferase activity in serum exosomes is an important biomarker for prostate cancer [35]. Importantly, EVs originating from LN18 cells have a higher concentration of several amino acids (glycine, threonine), as well as homoserine which is an intermediate in the biosynthesis of methionine, and threonine. 

Additional differences between cell lines in both cells’ and EVs’ metabolic profiles are shown in Figure 5. It is interesting to observe that GBMs and normal astrocytes (NHA), as well as different GBM cell lines, show higher significant metabolite differences in the EVs than in the cells, pointing to the great potential for EVs as diagnostic vesicles, keeping in mind the need to determine appropriate biomarkers for EV-based diagnostics as there is a significant difference between the metabolite profiles of EVs and their parental cells. 

Major metabolic differences in EVs, cells and Media of U118 and LN18 cells were determined using ANOVA running in Orange and shown in Figure 6A. Metabolomic measurements of EVs and media also allowed the analysis of correlation differences between different cell sources for metabolites. Figure 6B shows, as a heat plots, differences between correlation coefficients between metabolite concentrations in EVs and in media for U118 and LN18 cells, where red indicates a higher correlation in U118 and green, a higher correlation in LN18 cells. 

Protein measurements for EV-derived from U118 and LN18 cell lines have been previously published and made available by Lane et al. [15] with an extensive analysis of differences between LN18 and U118 cell lines and their EVs. Metabolic pathways that include a significant number of metabolites found to have significantly different concentrations in LN18 and U118 EVs (Figure 6), and proteins found to be present in LN18 but not U118 [15], are presented in Figure 7. The determination of statistically significant pathway enrichment was done using Metaboanalyst [36]. A significant overlap in pathways enriched with metabolic products and enzymes suggests a possibility for the enzymatic and metabolic functionality of EVs in the media. 

Further analysis of possible fluxes through metabolic reactions in enzymes was explored using Genome-scale metabolic modeling (GEM). GEM is one of the major system level modeling approaches describing a whole set of stoichiometry-based, mass-balanced metabolic reactions in an organism. GEM allows the prediction of metabolic flux values for an entire set of metabolic reactions using optimization techniques. Additionally, omics data can be used to calculate optimal flux rate through reactions in the system [37]. For human cells, the Recon 2 model is the most comprehensive metabolic network model, and currently includes 3288 genes, and 13,543 metabolic reactions involving 4140 unique metabolites [38]. Reactions possible in the two sets of EVs based on the previously published proteomics data [15] have been mapped onto Recon 3D metabolic network [38] in order to outline a subset of metabolic reactions possible in exosomes. Thus, based on proteomics data, possible metabolic reactions in EVs from U118 and LN18 cell lines are presented visually on the Recon metabolic network (Appendix A). These include the transport of metabolites across the EV membrane, as well as the number of metabolic reactions involving fatty acids, amino acids, sugars, etc. The mapping of possible pathways on the human cell metabolism network Recon (shown in Appendix A) includes compartmentalized metabolic processes. As there are no known compartments within EVs, the reaction flow would not be hampered by intracellular membrane metabolite localization.

The differences in the possible reactions based on protein content in LN18 and U118 cells are apparent, with major differences in fatty acid metabolism. Carnitine and glycerol are by-products of a number of fatty acid oxidation reactions and are both significantly over-concentrated in EVs from LN18 cells. Additionally, different protein content is apparent in the proteins involved in the urea cycle. Further differences are observed in the glycine metabolism, with the reaction involved in the production of glycerol from galactosyl–glycerol only possible in LN18 EVs. 

The possible enzymatic reactions in EVs originating from U118 and LN18 cells can be further explored by optimizing reaction fluxes using algorithms available within COBRA for the reconstruction of context-specific networks from omics data, such as GIMME [39]. Flux optimization provides hypothetical fluxes through all Recon 3D reactions based on the information about the measured proteins. 

The GIMME method minimizes the utilization of low expression reactions while keeping the model’s objective (here set to biomass maintenance) above some threshold value. In this application, the expression levels for proteins that were not observed in the proteomics measurement of EVs were set to 0 and observed protein levels were set to 100, having a boolean system of protein expression and optimization method searches for the flux model with the minimal use of low expression reactions.

The pathway, represented by the largest number of significantly differently concentrated metabolite and proteins found in EVs from LN18 but not in U118 EVs, is the citrate (TCA) cycle (Figure 7). The GIMME optimization of flux in LN18 and U118 EVs based on the protein presence for reactions grouped within TCA is shown in Figure 8. Based on the flux prediction, U118 cells have larger flux towards the production of succinate (succ) and no reactions for its further processing. In LN18 EVs, there is a significant flux towards malate (mal) from succinate. Once again, malate is seen as highly over-concentrated in A172 and, to a lesser extent, LN18 EVs relative to U118 EVs. 

Similarly, fluxes in EVs related to the glutathione metabolism pathway have a different, optimal route in EVs derived from U118 than LN18 cells in the production of oxidized glutathione (GSH) with a much larger flux in LN18-derived EVs for GSSG production (Figure 9). At the same time, fluxes in the synthesis of 5-oxoproline are higher in U118 cells. This simulation-based observation is in agreement with the measured metabolic profiles showing significantly higher concentrations of GSSG in LN18 EVs and 5-oxoproline in U118 EVs as well as in media (Figure 10).

## 3. Materials and Methods 

### 3.1. Cell Lines and Cell Culture

In this study, we used three different human human glioma cell lines: U118 (classified as grade IV as of 2007, glioblastoma; astrocytoma), LN-18 (classified as grade IV, glioblastoma; glioma) and A172 (classified as glioblastoma) from ATCC (diseases classification provided by ATCC), isolated small extracellular vesicles (EVs), and analyzed their metabolite content using NMR spectroscopy. GBM cells were grown in DMEM containing high glucose (Gibco laboratories, Gaithersburg, MD, USA) and supplemented with 10% Fetal Bovine Serum (FBS, Hyclone). Normal human astrocytes (NHA, Lonza, Walkersville, MD, USA) were used as normal counterparts and were grown in astrocyte medium-containing supplements (Allcells) and 5% FBS. All cells were grown in a humidified atmosphere of 5% CO_2_/95% O_2_ at 37 °C. The production of extracellular vesicles was done in serum-free conditions since FBS contains cow exosomes. Typically, cells were grown in T-175 flasks, washed 3x with PBS and cultured in 25 mL of serum-free medium for 72 h for accumulation of a sufficient number of EVs. 

### 3.2. Collection of Cells, Medium and Small Extracellular Vesicles

Small EVs or exosomes were isolated using a modified differential ultracentrifugation protocol that eliminates cell debris and microvesicles before pelleting small EVs or exosome-like vesicles [40,41]. Briefly, the media from 2–3 flasks from each GBM cell type and NHA cells were pooled and centrifuged at 700× *g* for 10 min at 4 °C to remove any floating cells, followed by centrifugation at 2400× *g* for 10 min at 4 °C. Cells were harvested by scraping in 10 mL PBS and pelleted by centrifugation at 300× *g* for 5 min at 4 °C. The supernatants were filtered through 0.22 µm filters using the 50 mL vacuum filtration system (Steriflip, Millipore). At this point, an aliquot of 0.5 mL medium from each cell line was frozen at −80 °C for metabolite extraction. The clarified conditioned media were transferred to ultracentrifuge tubes and centrifuged at 100,000× *g* for 70 min at 4 °C in Optima XPN-100 ultracentrifuge with 60 Ti rotor (Beckman Coulter, Mississauga, ON, Canada). The supernatants were decanted and the pellets containing small EVs were resuspended in 1 mL PBS, transferred to new ultracentrifuge tubes and centrifuged again at 100,000× *g* for 70 min at 4 °C in Optima Max-XP Ultracentrifuge (Beckman Coulter, Mississauga, ON, Canada). The final pellet was collected for metabolite and protein extractions. 

### 3.3. Western Blots

Cells and exosomes were lysed in Radioimmunoprecipitation assay (RIPA) buffer (Sigma-Aldrich, Oakville, ON. Canada) and quantified using the Bradford assay. Proteins were extracted by boiling for 5–10 min in Laemmli buffer (Biorad, Hercules, CA, USA) containing freshly added beta-mercaptoethanol (Sigma-Aldrich, Oakville, ON, Canada). Protein extracts were subjected to SDS-PAGE using Mini-PROTEAN precast gel 4%-15% (Biorad labs, Mississauga, ON, Canada), stained with Ponceau S and transferred to nitrocellulose membranes (0.45 µm, Amersham Protran). Membranes were blocked in 5% skim milk in TBST buffer (10 mM Tris, pH 7.4, 150 mM NaCl, 0.02% Tween-20) for 1 h followed by incubation with rabbit monoclonal anti-CD9 antibody diluted 1:2000 in the same buffer (Abcam, Toronto, ON, Canada) and incubated overnight at 4 °C. Membranes were washed 3x in TBST and then incubated with goat anti-rabbit-HRP antibody diluted at 1:5000 in TBST (Sigma-Aldrich, Oakville, ON, Canada) for 1 h at room temperature. Membranes were washed 3x with TBST, and then HRP signal was detected using Enhanced Chemiluminescent (ECL) reagent kit (Boster, Pleasanton, CA, USA).

### 3.4. Metabolite Extraction

Metabolites were extracted from GBM and NHA parental cells, media and EVs. Briefly, cell pellets were washed again with 10 mL PBS to remove any residual media and centrifuged at 300× *g* for 5 min at 4 °C. Cell pellets were held on ice for 5 min to slow down metabolism before being resuspended in 1 mL cold, chilled at −20 °C acetonitrile/water (1:1 *v*/*v*) mixture, which further quenches metabolism and lyses cells. Cell suspensions were then centrifuged at 12,000× *g* for 10 min at 4 °C. The supernatants were dried using a SpeedVac (Freeze dryer FTS Systems, FD-3-85A-MP) overnight at −80 °C. A similar protocol was followed for the extraction of intra-exosomal metabolites, except pellets were resuspended in a 0.2 mL cold acetonitrile/water mixture.

### 3.5. NMR Sample Preparation

For NMR sample preparation, D_2_O and Standard solution (NMR grade) was added for a total volume of 160 µL. Standard solution was added at 10% total sample and consisted of 50 mM sodium phosphate (pH 7.0), 0.5 mM sodium azide and 0.1% DSS. For dry samples, 160 µL D_2_O was mixed with 16 µL standard, and for liquid samples 100 µL medium was mixed with 60 µL D_2_O and 16 µL standard. Using gel loading tips, ~10 µL of sample was loaded into 3 mm Wilmad NMR tubes (Sigma-Aldrich, Oakville, ON, Canada) and subjected to NMR analysis.

### 3.6. NMR Experimentation, Data Processing and Quantification

All ^1^H NMR spectroscopy measurements were performed on a Bruker 600 MHz spectrometer at 298 K. One dimensional (1D) ^1^H (proton) NMR spectra were measured for all samples using 1D ^1^H with water suppression sequence (NOESY 1D). All spectra were processed using MestReNova 9.1.0 software (Mestrelab Research Solutions). Preprocessing for spectra included: exponential apodization (exp 1); global phase correction, as well as manual phase correction and baseline polynomial correction when needed; and normalization using the reference peak. Spectral regions from −0.5–10 ppm were included in the normalization and analysis. 

The assignment of peaks was performed using Madison Metabolomics Consortium Database and tools [42], HMDB [43] as well as NMR spectral peaks search tool MetaboHunter [25] and the literature assignments for metabolites previously observed in related samples. A total of 50 metabolites were included in the analyses. Spectra for metabolites that were used in the quantification of 1D ^1^H spectra were obtained from the Human Metabolomics Database (www.hmdb.ca) or Biological Magnetic Resonance Databank (www.bmrb.wisc.edu) and processed using MestReNova 9.1.0 software. Spectral preprocessing for standards spectra included: exponential apodization (exp 1); global phase correction; and normalization using the total spectral area. Spectral regions from −0.5–10 ppm were included in the normalization and analysis. Prior to quantification analysis, the standard spectra were aligned to the reference peak (DSS) using peak alignment by fast Fourier transform cross-correlation [44]. 

An automated method for quantification, using multivariable linear regression that finds the best fit of spectra for individual metabolites from database to the measured 1D sample spectra, was developed previously [45] and utilized in this study to determine relative metabolite concentrations. The partial least square regression analysis result was used as the starting point and the model was constrained to concentrations greater than or equal to zero. The deconvolution of spectra of mixtures, such as in metabolomics, with many strongly overlapping lines, possibly with an unknown number of lines and atomic groups, each with a different line width, is extremely difficult, and thus it is important to determine an optimal solver for this problem. The best result, i.e., the model with a minimal error was obtained with Levenberg–Marquardt curve fitting and this method was used for the quantification of metabolic data used in further analysis. Multivariate linear regression analysis was performed using lsqcurvefit running under Matlab. Metabolite concentrations across samples were determined using the same standard spectra that were normalized to a total intensity equal to one and sample spectra normalized to the same reference concentration. The resulting concentrations, therefore, provide relative metabolite measures in different samples using the same standard scale, allowing for comparison between samples without requiring absolute metabolite concentrations.

### 3.7. Data Analysis and Metabolism Modeling

Pre-processing, including data organization, the removal of undesired areas, normalization, as well as data presentation, was performed with Matlab R2019a (Mathworks). Minor adjustments in peak positions (alignment) between different samples were performed using Icoshift [22]. Principal component analysis (PCA), performed in Matlab using routine pca and ppca for probabilistic principal component analysis [46], was performed on sample spectra as well as relative metabolite concentration data. T-distributed stochastic neighbor embedding (t-SNE) [23] was performed in Matlab using function tsne. The selection of metabolic panels with statistically significant difference between groups was done using Orange, a component-based data-mining software running under Anaconda Python Data Science Platform (https://anaconda.org/; https://orange.biolab.si/) were used for feature selection. Specifically, feature selection was done using Logistic Ridge Regression performing L2 regularization as well as ANOVA ranking, as presented in Orange.

The modeling of metabolic flux was performed using COBRA running under Matlab (https://opencobra.github.io/cobratoolbox/stable/) with the reconstruction of context-specific networks from omics data performed using GIMME [32]. The GIMME method minimizes the utilization of low expression reactions while keeping the model’s objective (here set to biomass maintenance) above some threshold value. Recon3D [38], currently the most comprehensive, manually curated, genome-scale reconstruction of human metabolism, was downloaded from http://bigg.ucsd.edu/models/Recon3D and used for COBRA modeling. Recon3D includes 2248 open reading frames, 5835 metabolites, as well as 10,600 biochemical and transport reactions. The model is prepared for analysis using a procedure previously established by R. Fleming and provided at GITHUB [47]. Proteins present in EVs originating from U118 and LN18 were obtained from [15] and included for LN18 EVs 152 and U118 EVs 109 genes present in Recon3D model. Specific reactions for these genes were determined using mapGeneToRxn routine running under COBRA in Matlab, where the value for the measured genes was set to 100 and for genes that were not observed in [15] was set to zero. These reaction lists were used for GIMME optimization using a gurobi solver providing a model with minimal utilization of reactions with low expression value (here 0) and maximization of the use of high expression reactions (here 100) while maximizing flux in the biomass reaction. The aim of this model is to explore differences between possible metabolic processes in EVs originating from LN18 and U118 based on published proteomics information. Further experimental flux analysis and proteomics is required for the development of an actual model of EV metabolism.

## 4. Conclusions

The study of metabolomics as well as the molecular profiles of EVs are rapidly growing fields. Our metabolomics analysis of GBM cells, EVs and media confirms previously observed metabolic differences between GBM cells when compared to normal cells, with further differences in metabolic profiles in cells and EVs of GBM subtypes. The metabolic profiles of U118 cells were different from A172 and LN18 cell lines. Interestingly, metabolic contents of U118 EVs is much more significantly distinct from EVs of A172 and LN18. Major differences between U118 EVs and EVs from A172 and LN18 cells were observed both in qualitative PCA and t-SNE analysis as well as in the statistical analysis of quantified metabolic data. Additionally, in A172 and LN18, the metabolic profiles of EVs and cells showed a significant difference in the number of metabolites. In order to explore the possibility that the observed metabolic profile differences result from an active metabolism in EVs, we have used previously published proteomics data for EVs of LN18 and U118 cell lines [13] as representatives of these two groups, then simulated possible flux through the metabolic network in the interior of EVs. The results obtained from the flux simulations agree overall with the experimentally observed metabolic differences between EVs from different sources, leading us to hypothesize that EVs can have a functional metabolism, thereby changing their metabolic content. The possibility of their actively changing metabolome has major relevance for their application as diagnostic carriers. The biomarker panels or diagnostic models for EVs would have to be determined in the context of media (e.g., blood or CSF) and sample collection procedures in order to have appropriate biomarker panels in a metabolically active system. Furthermore, the utilization of EV-based metabolites as biomarkers for GBM requires further analysis in patient samples. Further experimental and computational analysis is underway in order to develop a detailed model of EV metabolism and to determine whether changes in their metabolic cargoes can have any significant role in GBM progression.

## Figures and Tables

**Figure 1 metabolites-10-00088-f001:**
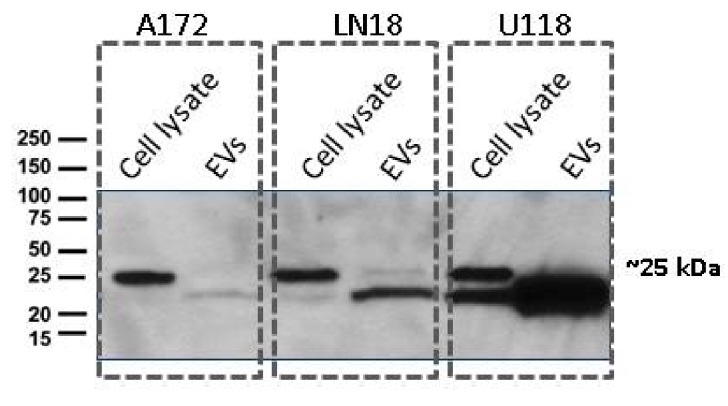
Cell and extracellular vesicles (EV) pellets were re-suspended in a Radioimmunoprecipitation assay (RIPA) buffer and subjected to SDS-PAGE and Western blot analysis. Samples were transferred to nitrocellulose membranes and probed with anti-CD9 antibody (1:2000, Abcam) followed by detection using goat anti-rabbit IgG-HRP conjugated secondary antibody. The signals were detected using an enhanced chemiluminescent (ECL) kit.

**Figure 2 metabolites-10-00088-f002:**
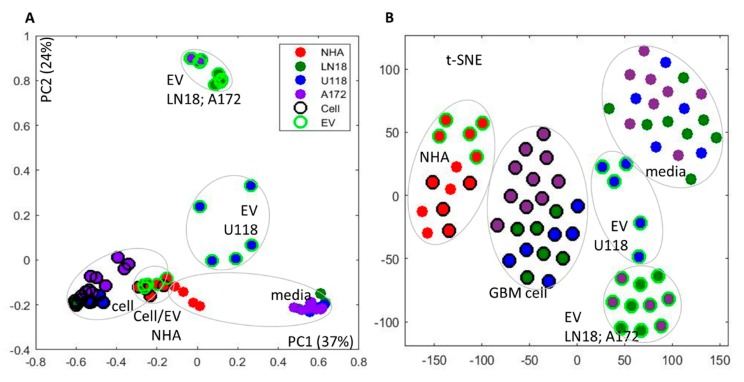
Analysis of sample differences from 1D Nuclear Overhauser Effect Spectroscopy (NOESY 1D) ^1^H NMR spectra of cells, media and EVs for glioblastoma (GBM) cell lines (LN18, A172 and U118) and normal human astrocytes (NHA). (**A**) Principal component analysis (PCA); (**B**) t-distributed stochastic neighbor embedding (t-SNE). Grouping of sample source (cells, EV, media) as well as cell type is indicated.

**Figure 3 metabolites-10-00088-f003:**
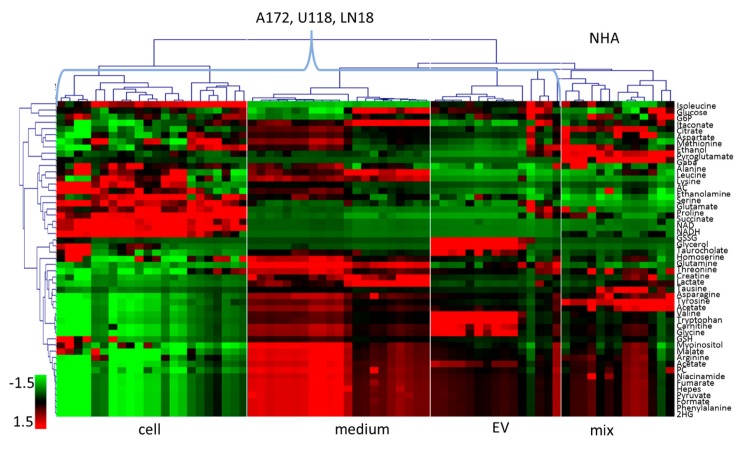
Relative concentrations of metabolites determined and quantified from NMR spectra. Metabolites are ordered using hierarchical cluster analysis across all samples. Values are scaled to mean of 0 and standard deviation of 1 across all samples and metabolites. AC—Adenosylhomocysteine; 2HG —2-hydroxyglutarate; GSSG—Glutathione (oxidized); GSH—Glutathione (reduced); G6P—Glucose 6-phosphate; PC—Phosphorylcholine.

**Figure 4 metabolites-10-00088-f004:**
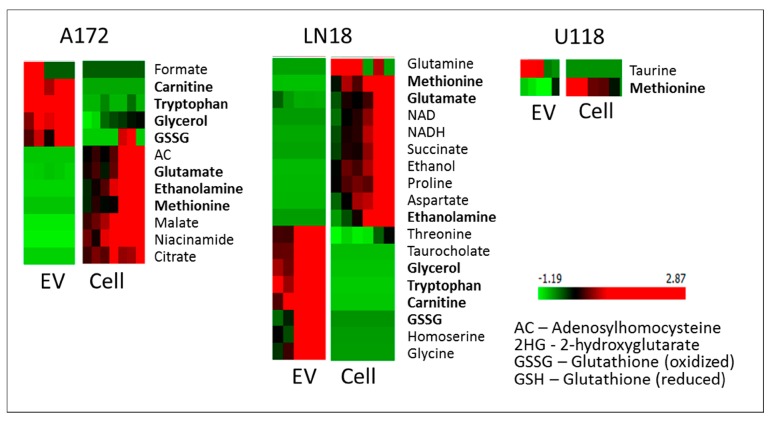
ANOVA selection of the most different metabolites between cell and EV extracts for GBM cell lines. Shown are metabolites with ANOVA > 5.

**Figure 5 metabolites-10-00088-f005:**
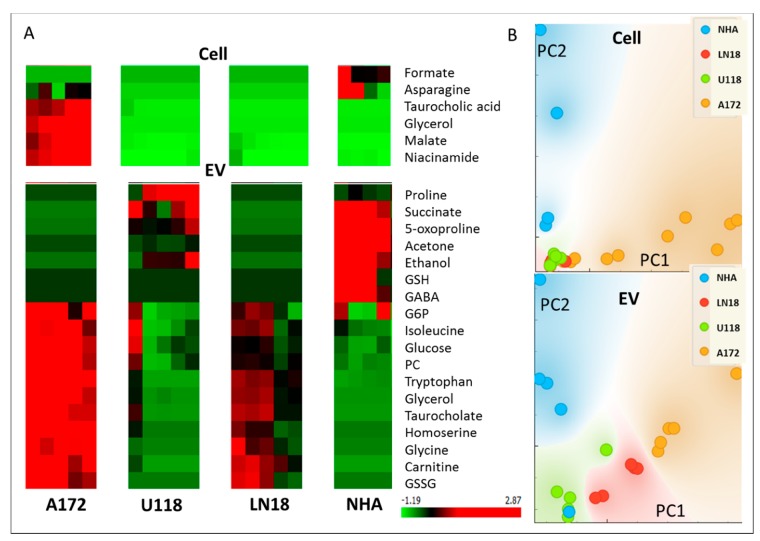
(**A**) ANOVA selection of the most different metabolites between cell four cell types in cells and EV extracts. Shown are metabolites with ANOVA > 5.G6P—Glucose 6-phosphate; PC—Phosphorylcholine; GSSG—Glutathione (oxidized); GSH—Glutathione (reduced). (**B**) PCA representation of sample groups obtained from the most significantly different metabolites in four sample groups for cells and EVs, presented in A.

**Figure 6 metabolites-10-00088-f006:**
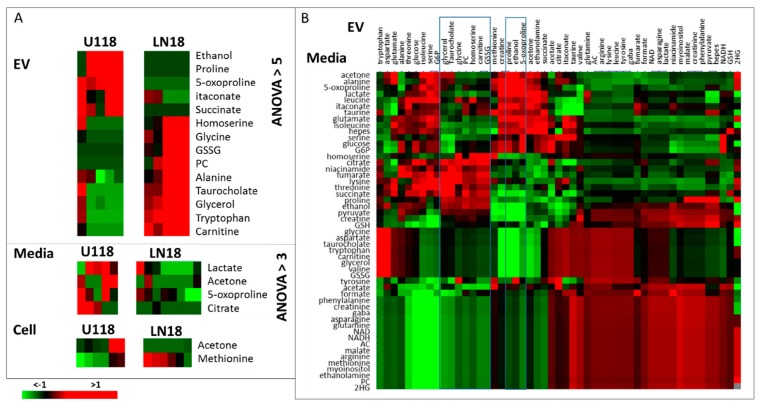
(**A**) ANOVA analysis of major metabolic differences between EVs, media and cells for LN18 and U118 lines. EVs show the biggest difference between two cell lines with the number of metabolites having an ANOVA value over 5. For cells and media, differences are much more subtle, with only a small number of metabolite showing an ANOVA value over 3. (**B**) The difference between correlation coefficients of metabolites in EV and media of U118 and LN18 cells, where a positive value (red) indicates a higher correlation in U118 cell lines and a negative value (green) shows a higher correlation in LN18 cells.

**Figure 7 metabolites-10-00088-f007:**
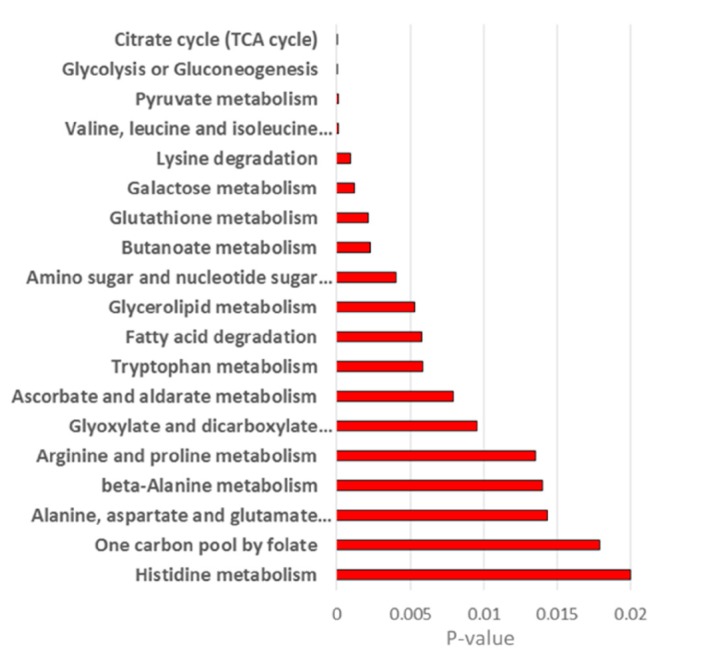
Metabolic pathways with the most significant enrichment for proteins found in EVs from LN18 but not EVs from U118 cells and metabolites showing different concentration between these two groups of EVs (Figure 6).

**Figure 8 metabolites-10-00088-f008:**
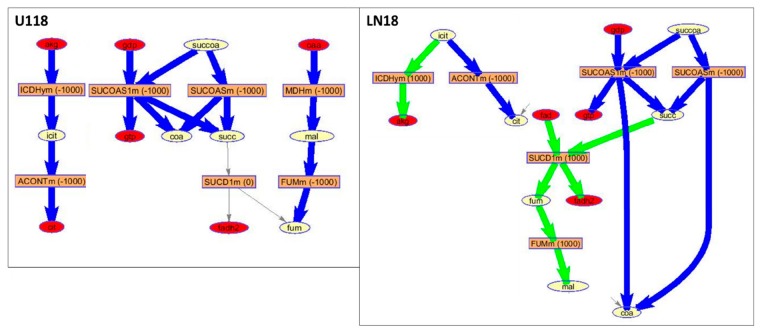
Part of the TCA-cycle-related metabolic processes. GIMME calculated fluxes in LN18 and U118 EVs made possible based on previously determined proteins in these vesicles. GIMME analysis provides a flux model with a minimized use of low-expression reactions while maximizing the objective reaction, in this case biomass preservation. Reactions are shown using the modeldraw.rxns routine in COBRA running under Matlab. Excluded from the representation are the cofactors including CO_2_, H_2_O, ATP, ADP, NAD, NADH, NADPH, NADP, H, Pi. In the figure, rectangles represent reactions with rates of fluxes in parentheses; ellipses represent metabolites; the red ellipses represent dead-end metabolites; gray arrows represent zero-rate fluxes; green arrows represent positive-rate (forward) fluxes; and blue arrows represent negative-rate (backward) fluxes. Reactions and metabolites notation is based on the Recon 3 metabolic network and is: akg—oxoglutarate (a-ketoglutarate), icit—isocitrate, cit—citrate, succ—succinate, mal—malate, fum—fumarate, coa—coenzyme A; gtp—guanosine triphosphate; fadh2—Flavin adenine dinucleotide.

**Figure 9 metabolites-10-00088-f009:**
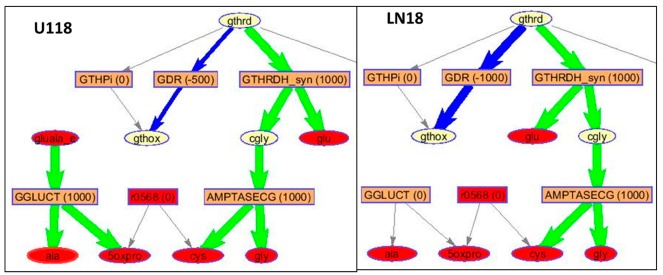
Part of the Glutathione metabolism flux GIMME optimization of flux through remaining Recon 3 reactions based on the present proteins in EVs. Reactions are shown using modeldraw.rxns routine in COBRA running under Matlab. Excluded from the representation are the cofactors including CO_2_, H_2_O, ATP, ADP, NAD, NADH, NADPH, NADP, H, Pi. Rectangles represent reactions with rates of fluxes in parentheses; the red rectangles represent reactions with only one metabolite; ellipses represent metabolites; the red ellipses represent dead-end metabolites; gray arrows represent zero-rate fluxes; green arrows represent positive-rate (forward) fluxes; and blue arrows represent negative-rate (backward) fluxes. Reactions and metabolites notation is based on the Recon 3 metabolic network and is: gthrd—reduced glutathione; gthox—oxidized glutathione; glu—glutamine; gly—glycine; ala—alanine; cys—cysteine; 5oxpro—5-oxoprolinate; cgly—carbamoyl glycine.

**Figure 10 metabolites-10-00088-f010:**
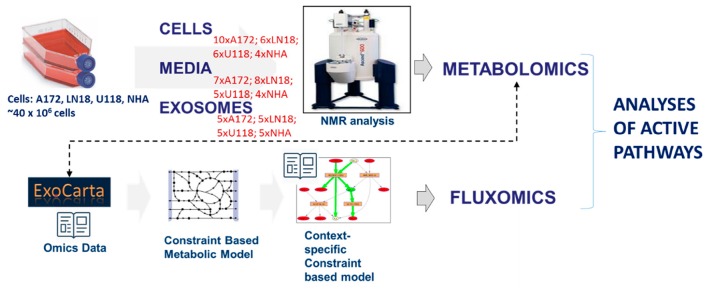
Experimental workflow of sample preparation for cells, media and EV analysis by NMR spectroscopy. Metabolomics, as well as computational flux simulation, allowed the investigation of possibly active pathways in EVs. Shown in red are the number of biological replicates measured for each cell and sample type.

**Table 1 metabolites-10-00088-t001:** Molecular classification of glioblastoma (GBM) cell lines according to several significant GBM markers. Obtained from [20,21] and www.expasy.org.

Cell Line	PTEN	PTEN WB	CDKN2A	EGFR	EGFRvIII	ExPASy Disease Assignment
U118	N	Yes	Del HOMO	G *	No *	Astrocytoma
LN18	N	No	Del HOMO	N *	No *	Glioblastoma
A172	del HOMO *	No	Del HOMO	G *	No *	Glioblastoma

* Del HOMO: homozygous deletion; N: No copy number change; No: EGFRvIII mutation not detected; G: Gain.

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
