# Peer review of "Analysis and Simulation of Glioblastoma Cell Lines-Derived Extracellular Vesicles Metabolome"

_metabolites, 2020, doi:10.3390/metabo10030088_

Round 1
Reviewer 1 Report
In this article, Cuperlovic-Culf et collaborators describe the glioblastoma-derived extracellular vesicles metabolome and metabolism by using multiple techniques. The authors found large differences in the metabolome of small EVs derived from GBM cells and compare to the metabolic profile of media, and to GBM cells.
This is a very interesting study, well conducted, with a massive amount of data. I recommend accepting this article after few changes as listed below.
Major points:
Were the authors analysing the different media without any cells ? As the media are basically different, are there clear differences ?
Minor points :
the figure 8 is not of good quality. Can the authors bring the high quality figure as supplemental material ? It is known that EVs contains more than metabolites. Can the auhtors discuss if active metabolite enzymes could be present in those ? It is partially discussed with figure 8, but a deeper discussion would be appreciated. Please check the text as there are very minor spelling mistakes.Author Response
We would like to thank Reviewer for valuable points. We have made all the requested change in the text and answers are provided below (following reviewer's comments in Italic).
Major points:
Were the authors analysing the different media without any cells ? As the media are basically different, are there clear differences ?
The same media formulation was used for 3 GBM cell lines. In order to grow NHA cells it was necessary to use different medium containing supplements for astrocytes. This is apparent in Figure 2 with clear separation of NHA media from the media of GBM cell lines. The major focus in this work is on the differences between GBM cell lines and therefore having the same media allowed us to make the comparison without bias. Major metabolic differences in the media of U118 and LN18 cells in shown in Figure 6.
Minor points :
the figure 8 is not of good quality. Can the authors bring the high quality figure as supplemental material ? It is known that EVs contains more than metabolites. Can the auhtors discuss if active metabolite enzymes could be present in those ? It is partially discussed with figure 8, but a deeper discussion would be appreciated. Please check the text as there are very minor spelling mistakes.
Figure 8 is now provided as supplementary material as well for more detailed investigation. The analysis of the possible function of previously determined enzymes in exosomes is discussed with specific analysis of possible pathways shown in Figure 7 and example of active pathways based on COBRA simulation presented in the text and Figure 9 and further discussed and included in the conclusions.
We have carefully read through the manuscript and corrected some mistakes in the text.
Reviewer 2 Report
The article is devoted to the important topic – finding of biomarkers for glioblastoma, and elucidation of metabolites and metabolic pathways involved in extracellular vehicles of the cancer cells.
The main questions:
What is the difference of the cell lines used? Do these have the different somatic and maybe also germline mutations profile defining such great differences in metabolism? If these are different – show what can lead to the different metabolism.
The description of COBRA and GIMME modeling is not sufficient. The reader needs much more details of this modeling including the proper figure legends describing all.
The lists of metabolites for all cell lines and extracellular vehicles need to be included to the article.
Do authors think that EV metabolites would correspond to the patients’ serum metabolites? If these are not, how the elucidated “biomarkers” can be used in clinic?
Author Response
We would like to thank reviewer for valuable comments. All requests have been no addressed and comments are below (following reviewer's points and represented in Italic):
The article is devoted to the important topic – finding of biomarkers for glioblastoma, and elucidation of metabolites and metabolic pathways involved in extracellular vehicles of the cancer cells.
The main questions:
What is the difference of the cell lines used? Do these have the different somatic and maybe also germline mutations profile defining such great differences in metabolism? If these are different – show what can lead to the different metabolism.
We have included a table that describes some examples of differences between three cell lines. Extensive analysis of differences between these cell lines has been published previously and we have now added few new references outlining this work.
The description of COBRA and GIMME modeling is not sufficient. The reader needs much more details of this modeling including the proper figure legends describing all.
We have enhanced figure legends as suggested and added more details on the use of COBRA and GIMME modeling.
The lists of metabolites for all cell lines and extracellular vehicles need to be included to the article.
Quantified metabolic measurements will be provided as a supplementary data and Figure 3 shows all metabolites and their relative concentrations in these groups.
Do authors think that EV metabolites would correspond to the patients’ serum metabolites? If these are not, how the elucidated “biomarkers” can be used in clinic?
In this paper our focus is on the development of EV extraction protocol, NMR metabolomics analysis and exploration of a possibility for active metabolic processes in EVs through simulation. Although metabolic differences found in EVs could be seen in clinical samples in order to have clinical biomarkers it will be necessary to measure metabolomics of EVs from a sufficient number of patients and determine biomarkers from this data. Metabolic differences found in this work are only showing that even in the very controlled case, with cells growing in the same media during the same time period we observe differences in metabolic profile and that there is a possibility for metabolic changes in the EVs. This result leads to hypothesize that there are metabolic differences in EVs from different GBM subtypes tested here but that extreme caution in sample preparation will be required for the development of clinically reliable markers.
Reviewer 3 Report
This manuscript by Cuperlovic-Culf et al. reports on the metabolome of extracellular vesicles obtained from cell culture medium of three glioblastoma cell lines and, for comparison, an astrocytoma line. The overall conclusion is the metabolic profile of the cell lines is different. There was no marker that was specific for glioblastoma cells. The manuscript is well written. However, the conclusions are limited since GBM-specific metabolic markers in vesicles could not be identified. .
Specific comments:
The title suggests that the study was performed with tumor material or GBM patients. Please modify the title. It should indicate that the study rests on glioblastoma cell lines. Also, what do authors mean with vesicle “metabolism”? The vesicle composition was studied, not the turnover, i.e. metabolism. Accepted predictive markers for GBM are IDH1mt and the amount of the DNA repair protein MGMT. If MGMT would be found in vesicles, the method could have some impact on treatment options. Have DNA repair proteins been found in extracellular vesicles (Fig. 7 refers only to metabolic enzymes)? Are the cell lines MGMT expressing? If not, an MGMT expressing GBM line (such as T98G) should be included. It is conceivable that extracellular vesicles exhibit a signature specific for GBM. This, however, requires the analysis of a large panel of GBM cell lines and comparison with other cancer and normal cell lines and tumors in situ. Can extracellular vesicles be observed in GBM tissue (in situ)? Do extracellular vesicles contain DNA that could be amplified by PCR and studied as to GBM-specific signature? Please add a short comment on this in Discussion.
Author Response
We would like to thank Reviewer for valuable comments. We have made all required changes with specific response shown below in Italic, following Reviewer's comments.
Specific comments:
The title suggests that the study was performed with tumor material or GBM patients. Please modify the title. It should indicate that the study rests on glioblastoma cell lines. Also, what do authors mean with vesicle “metabolism”? The vesicle composition was studied, not the turnover, i.e. metabolism. Accepted predictive markers for GBM are IDH1mt and the amount of the DNA repair protein MGMT. If MGMT would be found in vesicles, the method could have some impact on treatment options. Have DNA repair proteins been found in extracellular vesicles (Fig. 7 refers only to metabolic enzymes)? Are the cell lines MGMT expressing? If not, an MGMT expressing GBM line (such as T98G) should be included. It is conceivable that extracellular vesicles exhibit a signature specific for GBM. This, however, requires the analysis of a large panel of GBM cell lines and comparison with other cancer and normal cell lines and tumors in situ. Can extracellular vesicles be observed in GBM tissue (in situ)? Do extracellular vesicles contain DNA that could be amplified by PCR and studied as to GBM-specific signature? Please add a short comment on this in Discussion.
We have changed the title to match reviewer’s request. Within this publication we are focusing on the metabolome of EVs and simulation of possible metabolic processes based on the present enzymes. Analysis of other biomolecules in EVs have been done in other publications and is not focus here. We have now included a comment about this in the manuscript.
Reviewer 4 Report
I think that this issue is very interesting. There are a lot of technologies and analises.
However, the manuscript needs some corrections and clarifications:
1) the manuscript is not according with new CNS 2016 classification, the introduction should be actualised.
2) figures quality should be improved and simplified. there is a lot information in each figure difficult to follow.
3) To highlight the main metabolites of this work that should be analysed in patients samples.
Author Response
We would like to thank Reviewer for valuable comments. We have made all required changes with specific response shown below in Italic, following Reviewer's comments.
1) the manuscript is not according with new CNS 2016 classification, the introduction should be actualised.
Classification for the cell lines provided by ATCC is now included in Materials and Methods and a table presenting some major differences and classification for the three cell lines is included in the text.
2) figures quality should be improved and simplified. there is a lot information in each figure difficult to follow.
Figure 8 is now also provided as Supplementary material with some clean up for other figures included in the resubmission.
3) To highlight the main metabolites of this work that should be analysed in patients samples.
We have now added to the conclusion following statement in order to clarify need for patient sample analysis for biomarker discovery.
“Furthermore, the utilization of EV based metabolites as biomarkers for GBM requires further analysis in patient samples.”
Reviewer 5 Report
This interesting work explores the metabolome of extracellular vesicles (EV) of three established glioblastoma cell lines. The authors first show, that the metabolome differed between the cell lines. Furthermore, the metabolome differed between the cell lines and the respective EVs. Interestingly, it was shown, that the metabolome in the EVs is not static, but changes with time through the metabolic flux. This is of high practical interest for the application for the diagnostic detection of EVs in patients. One should expect that the metabolome changes depending on the location of sampling (blood vs. CSF) and other influencing variables.
The results presented here will be of high interest to many readers. All results are sound and the conclusions drawn are convincing. The linguistic style will be sufficient for publication after some minor changes. In summary, I strongly recommend publication without further changes needed.
Author Response
We would like to thank Reviewer for valuable comments. We have made all required changes with specific response shown below in Italic, following Reviewer's comments.
The results presented here will be of high interest to many readers. All results are sound and the conclusions drawn are convincing. The linguistic style will be sufficient for publication after some minor changes. In summary, I strongly recommend publication without further changes needed.
Thanks you very much for your highly encouraging review. We have read through the manuscript and made some small linguistic corrections.
Round 2
Reviewer 2 Report
The description of COBRA and GIMME modeling is absolutely not sufficient. The reader needs much more details of this modeling including the proper figure legends describing all.
Authors have to write step-by step their methods in COBRA and GIMME, Otherwise all that description is completely useless.
Which information a reader gets from Figure 8?
Author Response
We would like to thank reviewer for these comments. We have now included all requested changes shown in Track Changes in the manuscript and outlined below (in italic).
The description of COBRA and GIMME modeling is absolutely not sufficient. The reader needs much more details of this modeling including the proper figure legends describing all.
Authors have to write step-by step their methods in COBRA and GIMME, Otherwise all that description is completely useless.
We have significantly extended Material and Methods: Data analysis and metabolism modeling section providing specific information about the COBRA and GIMME modeling (page 16 line 436-450). Also, we have extended Figure legends for figures 8 and 9 including the information about the COBRA routine and procedure used for the preparation of this figure.
Which information a reader gets from Figure 8?
Figure 8 is removed from the text and moved to supplementary material.